# Pathogenesis of Hypertension in Metabolic Syndrome: The Role of Fructose and Salt

**DOI:** 10.3390/ijms24054294

**Published:** 2023-02-21

**Authors:** Manoocher Soleimani, Sharon Barone, Henry Luo, Kamyar Zahedi

**Affiliations:** 1Research Services, New Mexico Veterans Health Care Medical Center, Albuquerque, NM 87108, USA; 2Department of Medicine, University of New Mexico School of Medicine, Albuquerque, NM 87131, USA

**Keywords:** fructose, salt, small intestine, kidney tubules

## Abstract

Metabolic syndrome is manifested by visceral obesity, hypertension, glucose intolerance, hyperinsulinism, and dyslipidemia. According to the CDC, metabolic syndrome in the US has increased drastically since the 1960s leading to chronic diseases and rising healthcare costs. Hypertension is a key component of metabolic syndrome and is associated with an increase in morbidity and mortality due to stroke, cardiovascular ailments, and kidney disease. The pathogenesis of hypertension in metabolic syndrome, however, remains poorly understood. Metabolic syndrome results primarily from increased caloric intake and decreased physical activity. Epidemiologic studies show that an enhanced consumption of sugars, in the form of fructose and sucrose, correlates with the amplified prevalence of metabolic syndrome. Diets with a high fat content, in conjunction with elevated fructose and salt intake, accelerate the development of metabolic syndrome. This review article discusses the latest literature in the pathogenesis of hypertension in metabolic syndrome, with a specific emphasis on the role of fructose and its stimulatory effect on salt absorption in the small intestine and kidney tubules.

## 1. Introduction

Metabolic syndrome is a cluster of conditions manifested by visceral obesity, hypertension, glucose intolerance, hyperinsulinism, and atherogenic dyslipidemia [1,2,3,4]. Data from the Centers for Disease Control (CDC) indicate that 1 in 3 adults now fit the criteria for metabolic syndrome [5]. This escalation correlates with the introduction of high fructose corn syrup and elevated dietary salt intake that began in the late 1960s [6]. In addition, recent studies indicate that the onset of obesity, type 2 diabetes, and metabolic syndrome is on the rise in younger individuals and these are no longer conditions only affecting adults and the elderly [7,8,9,10,11]. Chronic diseases driven by metabolic syndrome accounted for approximately $543 billion in direct healthcare costs in the US in 2017 [12]. Hypertension is a major component of metabolic syndrome and is associated with a significant increase in premature morbidity and mortality due to stroke, cardiovascular ailment, and kidney disease [1,2,3,4]. Although the increase in the incidence of metabolic syndrome is attributed to increased caloric intake and decreased physical activity, the pathogenesis of hypertension in this disease remains poorly understood.

Factors that are associated with metabolic syndrome [13,14], and contribute to the development of hypertension include (1) obesity, (2) insulin resistance, (3) the Renin Angiotensin Aldosterone system (RAAS), (4) catecholamines, (5) oxidative stress, (6) inflammatory mediators, (7) sleep apnea, and (8) fructose and salt overload. The role of each of the above factors in metabolic syndrome will be discussed below.

***Obesity***. Visceral obesity has been proposed as an important player in the development of hypertension [15,16,17,18]. Published studies show that adipose tissue may function as a major endocrine organ by enhancing the secretion of various substances, or adipocytokines, which include leptin, tumor necrosis factor-α (TNF-α), interleukin-6 (IL-6), angiotensinogen, and non-esterified fatty acids (NEFA) [19]. Adipocytokines may contribute to the development of hypertension [19]. In support of this conclusion, it is worth mentioning that enhanced fructose consumption elicits the development of chronic low-grade inflammation and expansion of white adipose tissue in various models of metabolic syndrome [20], which may ultimately contribute to the generation of hypertension. 

The role of enhanced fructose and salt consumption in obesity and adipocytokine elaboration has been documented [20,21,22]. These studies demonstrate that adipocytokines, such as MCP-1, TNF-a, IL-6, and IL-17a, enhance renal sodium retention and alter vascular tonicity leading to the development of hypertension [20,21,22]. 

***Insulin resistance***. In addition to visceral obesity, insulin resistance is a major contributor towards the onset of metabolic syndrome and has been proposed to play an important role in the development of hypertension through several mechanisms [23,24,25,26,27,28,29,30,31]. Increased fructose intake is closely associated with the development of insulin resistance [32,33]. Several reports point to the anti-natriuretic effect of insulin (e.g., insulin-mediated enhanced salt absorption in the kidney) as a major mechanism responsible for the development of hypertension in metabolic syndrome [34]. It is noteworthy that the insulin resistance in metabolic syndrome presents with an increased concentration of circulating insulin. Given the fact that the anti-natriuretic effect of insulin is intact, this may lead to enhanced kidney salt absorption in individuals with insulin resistance [35,36,37]. In a large study on individuals with metabolic syndrome, the body mass index (BMI) and blood pressure were directly associated with the enhanced fractional absorption of sodium in the proximal tubule (PT) [38]. The highest age-adjusted levels of increased absorption of sodium in the PT were detected in obese hypertensive and obese insulin-resistant participants. There was also a direct association between uric acid and sodium absorption in the PT. The effect of insulin on augmented expression and activity of Na^+^-K^+^-ATPase activity in the PT has also been documented, which is consistent with enhanced salt absorption in the PT by insulin [39]. There are reports indicating that increased circulating insulin may be associated with greater epithelial sodium channel (ENaC) activity in the collecting duct (CD) [40]. This latter assertion, however, has been disputed [41]. There are also reports indicating that insulin increases sodium absorption in the kidney thick ascending limb of Henle (TAL) [42]. Overall, insulin resistance is proposed to be associated with the development of salt-sensitive hypertension through the anti-natriuretic effect of insulin [35,36,43]. 

The published data on the role of insulin on salt absorption in the PT point to insulin enhancing salt absorption via NHE3 working in tandem with the basolateral NBCe1, as well as the sodium/glucose symporter SGLT2 [44,45,46,47,48]. However, the role of enhanced salt absorption in the distal nephron, including the sodium chloride co-transporter (NCC) in the distal convoluted tubule (DCT) or ENaC in the CD, by insulin needs further investigation [49,50]. In vivo studies conducted in insulin-resistant individuals have also demonstrated that high circulating insulin levels are associated with an increase in circulating levels of endothelin-1, a peptide produced by endothelial cells that has vasoconstrictor activity [51], which could contribute to the generation of hypertension in insulin resistance. Figure 1 is a schematic diagram demonstrating the stimulatory effect of insulin on salt transporters in the kidney proximal tubule, the thick ascending limb of Henle, the distal convoluted tubule, and the collecting duct. 

***Renin-Angiotensin-Aldosterone System (RAAS).*** The RAAS plays a crucial role in blood pressure regulation by enhancing salt absorption in the kidneys, as well as by modulating vascular tone. Overfeeding of rodents increases the adipocytes’ angiotensin II concentration [52]. Further, the production of angiotensin II in adipose tissue and circulating levels of aldosterone are increased in obese subjects [53]. Published reports indicate high fructose consumption activates RAAS, thus contributing to the generation hypertension. The improvement in BMI, either through diet or bariatric surgery, is associated with the reduction in RAAS activity and hypertension in obese individuals [54]. Angiotensin II may play additional roles in elevating blood pressure through the stimulation of RAS homology, Family A (RhoA) activity, and oxidative stress, which can inhibit Phosphatidylinositol 3-kinase/Protein Kinase B (PI3K/Akt) signaling, resulting in decreased nitrous oxide (NO) in endothelial cells while increasing vasoconstriction [34]. 

***Catecholamine, oxidative stress, inflammatory mediators and sleep apnea.*** Several studies have indicated a role for catecholamine (e.g., adrenalin and nor-adrenalin) over-activity as a causative factor in the generation of hypertension in combination with obesity. Serum catecholamine concentrations were significantly increased in obese individuals as compared with lean individuals [34,55].

Oxidative stress due to increased generation of reactive oxygen species could result from enhanced expression of NADPH-oxidase and super oxide dismutase in response to high sodium levels [56,57,58,59]. These states are magnified in metabolic syndrome and are linked with sodium retention and salt sensitivity [60]. Further, elevated levels of inflammatory mediators may play an important role in the pathogenesis of hypertension in metabolic syndrome [61,62], in part, through the induction of renal and vascular inflammation and injury [63,64,65,66].

Obstructive sleep apnea (OSA) that has a high prevalence in individuals with metabolic syndrome, and is associated with sympathetic overactivity and hypertension [67], can also contribute to elevated blood pressure.

Most of the above factors (elevated circulating insulin, renin angiotensin aldosterone system, oxidative stress, etc.) promote salt retention [24,68,69,70], which can contribute to the development of tubular and vascular damage and the generation of hypertension. 

**Association of Obesity with Hypertension in the Million Veteran Program (MVP).** Published reports have pointed to a strong association between hypertension and obesity [71,72], as well as dietary fructose and salt intake in metabolic syndrome [1,2,3]. The Million Veteran Program (MVP) is a database administered by the Veterans Administration Department that has collected clinical and genetic data from over 900,000 veterans and non-veteran individuals. We have analyzed the prevalence of hypertension and obesity in MVP database as will be discussed below. 

***MVP database analysis: incidence of hypertension and obesity***. We examined the presence of hypertension, obesity, or their combination in our veteran population using the MVP database [73,74]. Our analysis of the MVP data, which has the de-identified healthcare records of 802,621 veterans, shows that 68.6% of all participants (550,400/802,621) have hypertension (Figure 2A,B). These data were derived by performing a query using the ICD10 diagnostic criteria specific for hypertension, such as essential hypertension, hypertensive heart disease, hypertensive renal disease, unspecified secondary or pre-existing hypertension, etc. Our analysis further indicated that 45.62% have obesity. The population of veterans that suffer from both obesity and hypertension (Figure 2C) constitute 36.87% of individuals identified in our analysis and may represent those that have developed or are considered at risk for developing metabolic syndrome.

The data shows the gender and racial demographics in veterans (Figure 2A,B). As of 2022, the following age distributions were represented in the MVP database: (1) ages 18 to 49 made up 19% of the study; (2) ages 50 to 79 made up 73% of the study; and (3) ages 80 to 99 made up 9% of the study. The low number of female enrollees in the MVP database reflects the total number of female veterans in US. When using obesity-specific (BMI > 30) ICD10 codes in an MVP search, 275,200 of the participants (275,200/802,621 or 34.3%) were identified as obese, with an overwhelming proportion of those (222,600/802,621 or 27.7%) also displaying hypertension. It is highly plausible that the number of veterans diagnosed with obesity is underreported due to a failure to include obesity as a diagnosis, since BMI may not routinely be documented. Our search also revealed that 5400 veterans had undergone bariatric surgery (for BMI > 40) with an overwhelming number of those individuals (4300) also carrying the diagnosis of hypertension. The above results demonstrate that veterans, similar to the population at large, display a nearly epidemic proportion of obesity with hypertension.

**The Role of Enhanced Salt and Carbohydrate Consumption in the Pathogenesis of Hypertension in Metabolic Syndrome.** While age, sex, and age-related hormonal changes may contribute to the development of metabolic syndrome [75,76,77,78], at its core, this condition results from dietary habits (e.g., excess of energy due to increased caloric intake) and decreased physical activity [79,80]. Therefore, treating obesity is a good strategy to reverse the clinical features of metabolic syndrome. Both life modification approaches and bariatric surgery have shown good promise. 

Epidemiologic studies over the last three decades have shown that increased dietary sugar intake correlates with the soaring prevalence of metabolic syndrome. In contrast, individuals consuming healthier diets (e.g., Mediterranean Diet) may display reduced levels of obesity, hypertension, and incidence of metabolic syndrome [81,82,83,84,85,86]. According to the American Heart Association [87,88,89], Americans are consuming ~355 calories/day of sugar, mostly in the form of fructose (high-fructose corn syrup) and sucrose (a glucose/fructose disaccharide found in high abundance in many processed foods, such as candy, ice cream, breakfast cereals, canned foods, soda, and other sweetened beverages). In addition to elevated levels of dietary sugars, the CDC reports that Americans consume more than 2–3 times the recommended amount of salt, raising the risk of hypertension, cardiovascular disease, and kidney failure [87,88,89,90]. Published reports indicate that an enhanced consumption of fructose is crucial to the generation of hypertension in metabolic syndrome both in rodents [87,88,89,90] and humans [91,92,93,94]. 

***Pathways mediating carbohydrate and salt absorption in the small intestine.*** Carbohydrates, including glucose and fructose, are primarily absorbed in the small intestine via distinct transporters. Glucose is predominantly absorbed via sodium/glucose cotransporter 1 (SGLT1; SLC5A1) at the apical membrane, although glucose transporter 2 (GLUT2; SLC2A2) may also play a role. Fructose is primarily absorbed via GLUT5 [89,95]. Figure 3 is a schematic diagram depicting the localization of GLUT2, GLUT5, and SGLT1 in the small intestine. Although there is no cross-functional reactivity between GLUT5 and SGLT1, increased consumption of either fructose or glucose enhances the expression of both molecules [95,96,97]. 

Glucose-independent salt absorption in the small intestine is mediated via the chloride/base exchangers, down-regulated in adenoma (DRA; SLC26A3) and putative anion transporter 1 (PAT1; SLC26A6), working in parallel with the Na^+^/H^+^ exchanger 3 (NHE3) [98,99,100,101,102,103,104,105,106,107,108]. Figure 4 (left panel) is a schematic diagram depicting the localization of NHE3 and PAT1 in the small intestine. The basolateral Na^+^/K^+^ ATPase mediates the exit of sodium to the blood in the small intestine; Figure 4 (right panel) illustrates the apical localization of NHE3 and PAT1 on jejunal villi.

***Stimulatory effect of excessive fructose consumption on salt absorption in the small intestine.*** Given that increased dietary carbohydrate (fructose) and salt consumption is associated with hypertension, we asked whether there are close interactions between fructose and salt in the intestine and/or the kidney. Our laboratories were the first to show the identification of the glucose transporter isoform 5 (GLUT5 or SLC2A5) as the main fructose absorbing transporter in the small intestine [95]. Our studies demonstrated a dramatic impairment in fructose absorption in GLUT5 KO mice subjected to a high fructose (HF) diet [95]. Furthermore, we demonstrated that GLUT5 KO mice have a very low blood fructose concentration on either a HF diet or a normal chow [95]. These results strongly support the conclusion that GLUT5 is the dominant fructose absorbing transporter in the small intestine. Follow-up studies by other investigators have confirmed our findings that GLUT5 is the main fructose absorbing transporter in the small intestine [97]. 

Molecules mediating fructose-stimulated salt absorption in the small intestine. The emphasis on the pathogenesis of salt overload has predominantly focused on enhanced salt absorption in the kidney tubules. However, less attention has been paid to enhanced fructose-driven salt absorption in the intestine, a major contributor to salt overload in metabolic syndrome [109]. Salt overload occurs as a result of elevated salt intake and increased levels of osmolar sodium in the serum and interstitium [110,111]. A rise in salt intake can alter the osmolar sodium balance and lead to volume expansion and high blood pressure [110,111,112]. Another factor that needs to be considered is the presence of non-osmolar sodium in the skin and muscles that can act to buffer the changes caused by high sodium levels via its sequestration [112,113]. The mobilization or retention of non-osmolar sodium also plays an important role in the regulation of blood pressure [114,115,116]. 

Our laboratory was the first to identify the putative anion transporter (PAT1 or SLC26A6) as the main apical Cl^−^/HCO_3_^−^ exchanger in the small intestine [103,105,106]. Our studies indicated that PAT1 is predominantly expressed on the apical membrane of small intestinal villi, with very low expression in the large intestine [103,105,106]. In addition to PAT1, the other major apical Cl^−^/HCO_3_^−^ exchanger in the intestine is SLC26A3 (down-regulated in adenoma, or DRA). DRA is predominantly expressed in the large intestine and at lower levels in the small intestine where it plays a critical role in chloride absorption in both regions [99,102,104,107].

Given that the jejunum is the main location for the absorption of salt and fructose in the GI tract, and given the identical localization of GLUT5 and PAT1 (and NHE3) on the apical membrane of jejunum villi [109], we examined the effect of fructose on salt absorption and salt transporters in the small intestine. Our studies demonstrated that luminal fructose stimulated salt absorption in mouse perfused jejunum [95,109]. The expression levels of GLUT5, PAT1, and NHE3 significantly increased in the jejunum of mice subjected to an increased dietary fructose intake for 2 weeks [109]. The stimulatory effect of fructose on salt absorption was significantly blunted in PAT1 deficient mice [109]. 

Taken together, our results indicate that PAT1 is the target of fructose activation in the small intestine, which in collaboration with the Na^+^/H^+^ exchanger NHE3 mediates fructose-stimulated salt absorption in the intestine. In addition, we showed that fructose consumption impairs the excretion of salt by the kidney [109], leading to a state of salt overload even in the early stages of increased dietary fructose in mice and rats [109]. We further demonstrated that fructose-stimulated salt absorption in the intestine (via PAT1 and NHE3 working in tandem) and the impairment of salt excretion by the kidney play essential roles in the generation of fructose-induced hypertension [95,109]. This conclusion is based on studies showing the prevention of hypertension in PAT1 KO mice on HF diet [109]. 

A recent study showed a very robust enhancement in the expression of GLUT5 in the small intestine of obese subjects vs. lean individuals [117,118]. The remarkable activation of the fructose transporter GLUT5 in the small intestine of obese individuals [118] mimics the upregulation of GLUT5 in the jejunum of rodents fed high fructose diets [95,96,97,109], and points to the stimulatory role of increased fructose consumption on the expression of GLUT5 in humans with metabolic syndrome/obesity. 

The absorption of salt highlights the contribution of both chloride and sodium absorbing transporters working together. Published studies demonstrate that the Na^+^/H^+^ exchanger isoform 3 (NHE3) is the main sodium absorbing transporter in both the small and large intestines [98,100,101,108]. These studies also demonstrate that the contribution of the Na^+^/H^+^ exchangers NHE-2 and NHE-8 to salt absorption is minimal [99,102,103]. Similar to the small intestine, the kidney PT shows strong apical expression of NHE3 and PAT1, along with GLUT5 [101,104,107]. 

Given that the jejunum is the main location for the absorption of salt and fructose in the GI tract, and given the identical localization of GLUT5 and PAT1 on the apical membrane of jejunum villi (Figure 3 and Figure 4) and reference [109], we examined the effect of luminal fructose on salt absorption and salt transporters in the small intestine. Our studies demonstrated that luminal fructose stimulated salt absorption in mouse perfused jejunum [95,109]. 

The expression levels of GLUT5 and PAT1 significantly increased in the jejunum of mice subjected to an increased dietary fructose intake [109], and the stimulatory effect of fructose on salt absorption was significantly blunted in mice deficient in PAT1 [95,109]. The presence of GLUT5 is essential for the stimulatory effect of fructose on salt absorption, as luminal fructose failed to stimulate salt absorption in GLUT5-deficient mice [95]. Taken together, these studies demonstrate that fructose enhances salt absorption in the jejunum by stimulating NHE3 and PAT1, with GLUT5 being essential for this process [95,109].

***Effect of increased dietary fructose on renal salt excretion.*** The effect of increased dietary fructose on salt excretion in male Sprague Dawley rats has been examined [109]. Animals were placed in metabolic cages, and after acclimation, were subjected to balance studies. Daily urine collection and food intake measurements were performed for 5 consecutive days. Following the consumption of a normal diet for 24 h, animals were switched to a HF diet for 4 days. Food intake was comparable amongst animals on normal chow or on a HF diet. Results further indicated that daily urinary excretion of chloride and sodium in rats on a HF diet was significantly reduced, when compared with the same rats that were on a control diet before being switched to the HF diet (* *p* < 0.02, ** *p* < 0.005, and *** *p* < 0.002 vs. control diet) [109]. These results indicate that increased dietary fructose (and by inference sucrose) exerts two distinct effects on salt homeostasis: it enhances salt absorption in the small intestine and reduces salt excretion in the kidney. The net effect of these two alterations is the development of salt overload. 

Published studies clearly indicate that fructose stimulates salt absorption in the small intestine via the activation of PAT1 and NHE3 through a GLUT-5-dependent process. Recent studies by Gonzalez et. al. further implicate the important role played by SGLT5 in sodium coupled fructose transport in the renal proximal tubule epithelium [119]. In addition, fructose consumption impairs the excretion of salt by the kidneys. The effect of fructose on enhanced salt absorption in the kidneys involves specific transporters in multiple nephron segments, including the apical Na^+^/H^+^ exchanger, NHE3, SGLT5, and GLUT5 in the proximal tubule; the apical Na-K-2Cl cotransporter, NKCC2, in the thick ascending limb of Henle; and the apical Na-Cl cotransporter, NCC, in the distal convoluted tubules [120,121,122]. The schematic diagram in Figure 5 depicts the impact of increased dietary fructose on salt transporters in kidney tubules.

It is worth mentioning that long-term fructose consumption causes insulin resistance and increased circulating levels of insulin. The latter can contribute to enhanced salt absorption in the kidneys by activating NHE3, NKCC2, NCC, and the ENaC in the proximal tubule, the thick ascending limb of Henle, the distal convoluted tubule, and the collecting duct, respectively [35,36,38,39,40,41,42,43]. Enhanced absorption of salt is associated with increased expression of NADPH oxidase and superoxide dismutase, increased expression of reactive oxygen intermediates, and oxidative injury [56,57,58].

**Uric acid, high dietary fructose intake, and metabolic syndrome**. Fructose consumption raises uric acid production [123,124]. Recent data suggest that chronic hyperuricemia can play a role in the genesis of hypertension and metabolic syndrome in rodents, as well as in humans. Reductions in serum uric acid in fructose-fed rodents may attenuate several features of metabolic syndrome. Uric acid has the potential to accelerate renal disease in experimental animals and is associated with progressive renal disease in humans [123,124]. It is hypothesized that fructose-induced hyperuricemia may have a pathogenic role in metabolic syndrome, possibly due to its ability to inhibit endothelial function.

**Gut microbiome, obesity, and metabolic syndrome**. Associations between the gut microbiome, obesity, and metabolic syndrome have been investigated [4,125,126]. Recent insights have generated a new perspective suggesting that our microbiota might be involved in the development of these disorders [4,125,126]. Studies have demonstrated that obesity and metabolic syndrome may be associated with profound changes in the gut microbiome. Furthermore, a metabolic syndrome phenotype can be induced through fecal transplants, which corroborates the important role of the microbiota in this disease [4,125,126]. Dietary composition and enhanced caloric intake appear to swiftly affect intestinal microbial composition and function; therefore, more studies are needed to ascertain the exact role of altered gut microbiota in the genesis of obesity and metabolic syndrome.

**The Western Diet and metabolic syndrome.** In addition to high fructose and high salt, the Western diet contains an increased fat content. This combination of high fat, fructose, and salt intake has gained increasing attention with regard to the pathogenesis of metabolic syndrome due to it vigorous similarity to the dietary conditions responsible for other obesity-related conditions, such as nonalcoholic fatty liver disease (NAFLD) and nonalcoholic steatohepatitis (NASH) [127,128]. 

The increased consumption of carbohydrates (specifically fructose), along with fatty foods in rodents, causes obesity, elicits insulin resistance, exacerbates dyslipidemia [129], and elevates systemic blood pressure [130]. In addition, increased dietary fructose exacerbates high fat-induced hyperinsulinemia, fasting hyperglycemia, and hypertension [130]. In brief, the high-fat-high-fructose fed mice exhibited overt characteristics found in metabolic syndrome, including obesity, severe insulin resistance, dyslipidemia, significant hyperuricemia and hypertension [20,21,22,129,130]. 

## 2. Conclusions

Taken as a whole, a picture emerges where excessive fructose and salt intake can contribute to the development of multiple determinants of metabolic syndrome, including insulin resistance, low grade inflammation, renin angiotensin aldosterone system activation, elevated serum uric acid, and obesity. Coupled with the fructose-stimulated salt absorption in the small intestine and kidney tubules, these alterations will lead to a state of salt overload and eventual hypertension. Blocking or inhibiting the main carbohydrate-stimulated salt absorbing molecules in the small intestine (NHE3, PAT1, and SGLT) and kidney tubules (NHE3, NCC, and ENaC) may significantly blunt the generation of hypertension in individuals with metabolic syndrome. 

## Figures and Tables

**Figure 1 ijms-24-04294-f001:**
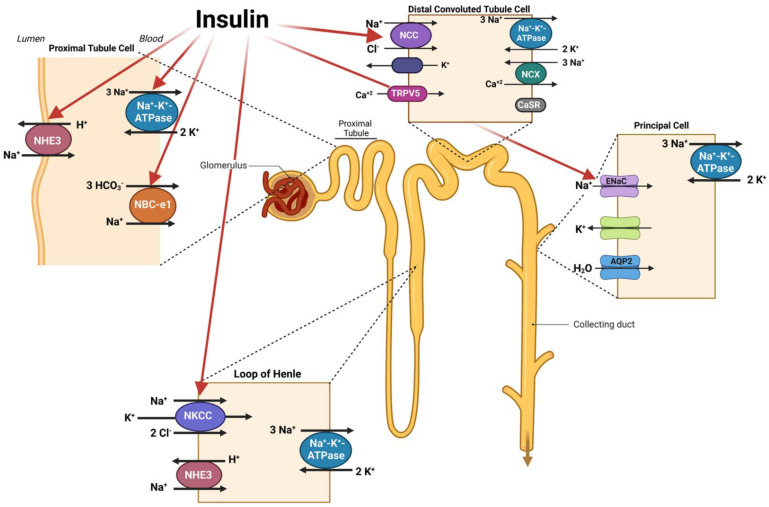
Schematic diagram depicting the stimulatory effect of insulin on salt transporters in the kidney tubules. The prevailing studies indicate that activity of multiple transporters in various nephron segments are upregulated in response to insulin and contribute to the enhanced transport of sodium. Created with Biorender.

**Figure 2 ijms-24-04294-f002:**
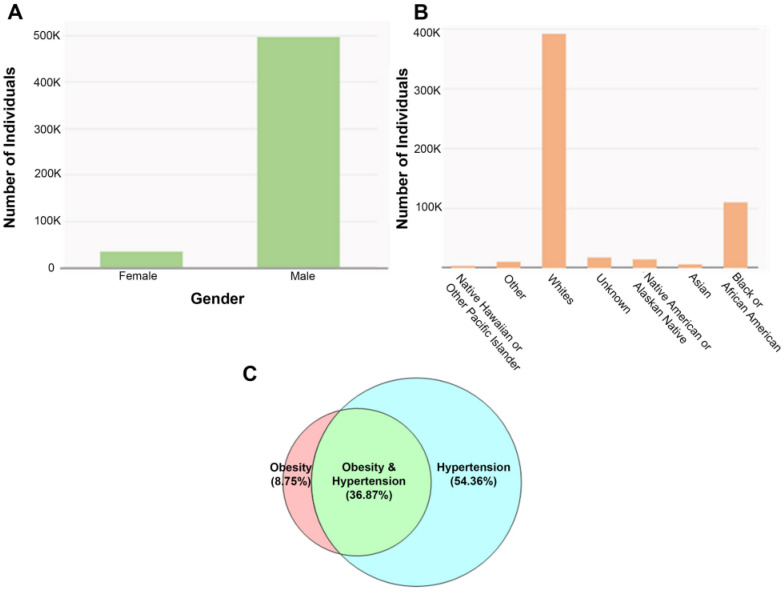
Prevalence of hypertension and obesity as extracted from MVP database. The number of veterans with hypertension based on (**A**) gender and (**B**) racial demographics. (**C**) A Venn diagram depicting the prevalence of hypertension, obesity, and both hypertension and obesity together in the veteran population.

**Figure 3 ijms-24-04294-f003:**
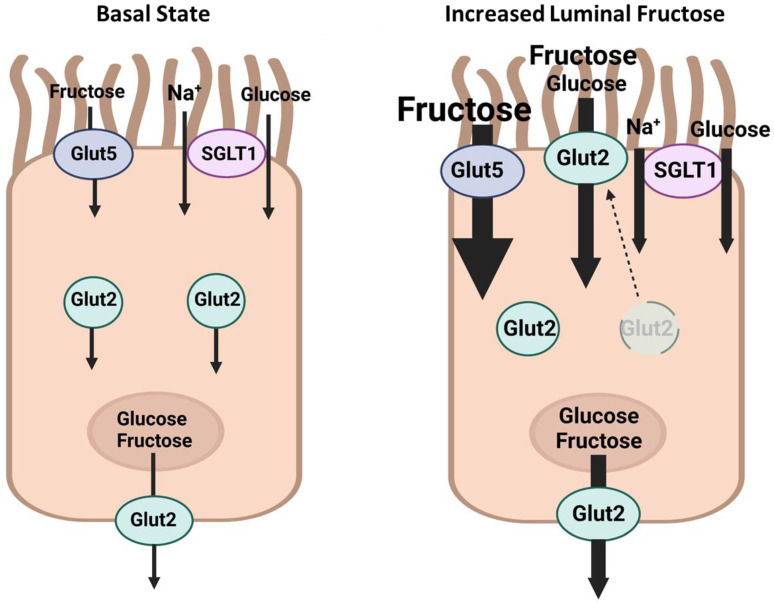
A schematic diagram demonstrating the localization of carbohydrate-absorbing transporters in the small intestine under basal state (left panel) and in the presence of increased luminal fructose (right panel). Glut5 (right panel) has a thickened arrow reflecting its enhanced expression in the presence of increased luminal fructose. SGLT1 (right panel) has a thickened arrow indicating its increased expression and the consequent increase in sodium and glucose transport in the setting of elevated luminal fructose. Glut 2 (right panel) is moved to the apical membrane showing an enhanced absorption of fructose and glucose in the presence of heightened luminal fructose. In addition to increased expression of Glut5 and SGLT1, Glut2 shows increased membrane targeting to the apical membrane in the presence of increased luminal fructose. Created with Biorender.

**Figure 4 ijms-24-04294-f004:**
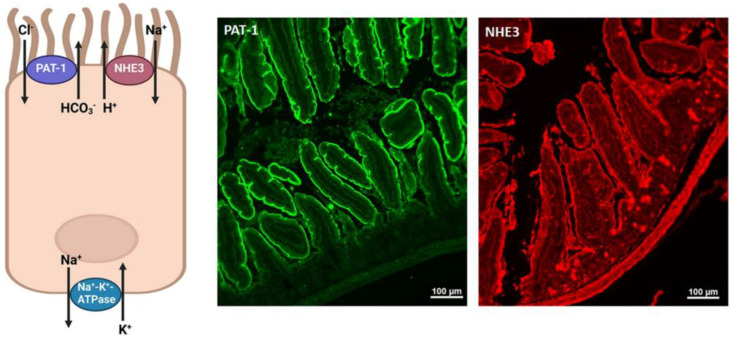
The localization of salt transporters, NHE3 and PAT1, in the small intestine (left, schematic diagram) and by immunofluorescence staining (right). Both NHE3 (red) and PAT-1 (green) show similar expression patterns on the apical membrane of jejunal villi (20× magnification; scale bar 100 μm). Schematic diagram created with Biorender.

**Figure 5 ijms-24-04294-f005:**
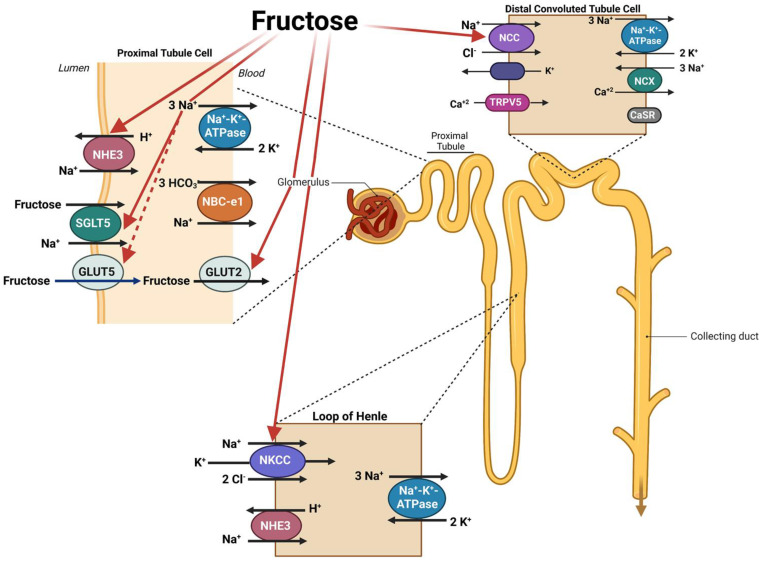
A schematic diagram depicting the stimulatory effect of fructose on salt transporters in the kidney tubules. Red arrows indicate kidney transporters that are affected by elevated levels of dietary fructose, including SGLT5 and GLUT5 which move luminal fructose into the proximal tubule and GLUT2 which acts to remove intracellular fructose from the cell. Created with Biorender.

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
