# Peer review of "Pathogenesis of Hypertension in Metabolic Syndrome: The Role of Fructose and Salt"

_ijms, 2023, doi:10.3390/ijms24054294_

Round 1
Author Response
The response to Reviewer 1 is attached

Reviewer 2 Report
Thank you for the opportunity to review this manuscript. The abstract lacks relevant background that supports the aim/purpose of this review. The authors need to be clear of the context for each point that they stated e.g. salt intake in metabolic syndrome or salt intake in relation to hypertension. The title or aim of the review is to examine role of fructose and salt in the pathogenesis of hypertension in metabolic syndrome. However, some sections of the manuscript focus on factors/mechanisms underlying the development of hypertension in metabolic syndrome- that have been discussed in other review papers (e.g. Yanai H, Tomono Y, Ito K, Furutani N, Yoshida H, Tada N. The underlying mechanisms for development of hypertension in the metabolic syndrome. Nutr J. 2008 Apr 17;7:10. doi: 10.1186/1475-2891-7-10. PMID: 18416854; PMCID: PMC2335113). It is difficult to understand why the authors want to focus on hypertension in metabolic syndrome as the manuscript seems to be discussing the effect of fructose and salt intake on hypertension itself. There are multiple grammatical errors throughout the manuscript and a language check is necessary. The authors should consider carefully what they want to discuss in this review e.g. fructose and salt in the pathogenesis of hypertension in metabolic syndrome or fructose and its stimulatory effect on salt intake on hypertension in metabolic syndrome.
Title: The authors should consider to revise the title to:
i) The Role of Fructose and Salt in the Pathogenesis of Hypertension in Metabolic Syndrome
ii) Pathogenesis of Hypertension in Metabolic Syndrome: The Role of Fructose and Salt
Abstract:
Lines 15-17: The authors stated that “According to the CDC, obesity in the US has increased drastically since the 1960s leading to chronic diseases and rising healthcare costs. Hypertension is a key component of metabolic syndrome/obesity”. However, the sentences seem inappropriate since the focus is on hypertension in metabolic syndrome. Obesity itself is a component or risk factor for metabolic syndrome and hypertension is a consequence of obesity-hypertension is not a component of obesity. Please clarify and focus on hypertension in metabolic syndrome.
Lines 21-22: Please remove “during the last four decades”.
Line 24: Is the manuscript focus on the US population or western diet? If no, the authors should avoid sentences that focus on Americans or western diet in particular. General sentences that take into account the overall population will be better.
Lines 27-28: The authors stated that “specific emphasis on the role of fructose and its stimulatory effect on salt absorption in the small intestine and kidney tubules”. However, the brief background in the abstract only mentioned intake of sugar and salt, and not on the role of fructose (as a sugar) in stimulating salt reabsorption. Additionally, this sentence contradicts the title that stated the role of fructose and salt on hypertension in metabolic syndrome and not the role of fructose and its stimulatory effect on salt reabsorption on hypertension in metabolic syndrome.
Introduction:
Line 31: Please only state “Introduction”.
Lines 33-41: Why is obesity being highlighted? Obesity and hypertension are criteria/components of metabolic syndrome, obesity is not a component of hypertension but obesity is a risk factor for hypertension. The authors need to clarify the link/relationship that they want to make. It is unclear how this part is related to hypertension in metabolic syndrome.
Lines 48-52: Irrelevant to the context of role of fructose and salt on hypertension in metabolic syndrome
Lines 53-110: These sections focus on the mechanisms underlying the development of hypertension in metabolic syndrome, and there are very little discussion on salt/salt retention and no discussion on fructose.
Lines 111-135: The authors presented evidence from epidemiological studies to support the well-established association between obesity and hypertension, there were no connection to salt and fructose intake/effect on hypertension in metabolic syndrome. Figure 2 is reflecting the discussion in lines 111-135 which is irrelevant to the aim of this review.
Lines 141-142: The subtitle “The Role of Excessive Carbohydrate Consumption in the Generation of Metabolic Syndrome and Hypertension” did not include salt consumption.
Figure 3 is not informative, the increased expression of GLUT5 and SGLT1 due to consumption of fructose is not well illustrated in the figure.
Lines 181-188: Please remove the repetitive figure legend for Figure 4.
Lines 193-293: This part focus on fructose-stimulated salt absorption/reabsorption, not on salt itself in the pathogenesis of hypertension in metabolic syndrome.
Line 199: Please change “GLU5” to “GLUT5”.
Lines 302-324: The authors provide additional information on other factors e.g. gut microbiome, uric acid, western diet that are associated with metabolic syndrome, but these are irrelevant to the context of fructose and salt in the pathogenesis of hypertension in metabolic syndrome.
Lines 325-332: This should be under a Section title “Conclusion”
Author Response
The response to Reviewer 2 is attached

Reviewer 3 Report
The comments and recommendations for the authors are in the attached file below.

Author Response
The response to Reviewer 3 is attached.

Round 2
Reviewer 2 Report
Thank you for the opportunity to review the revised manuscript. The authors have made some amendments. However, most of the responses did not directly address the reviewer’s main concerns. The structure of the manuscript remains disorganized and it lacks details and description on important points.
General comments: It remains unclear if the focus of the review is on role of fructose and salt on hypertension in metabolic syndrome or is it on “the role of fructose in stimulating salt absorption and its impact on the development of hypertension in metabolic syndrome” as given as response to the reviewer’s comment. If it is the latter, please kindly revise the title and structure of the manuscript accordingly.
Line 46: Please change “this condition” to “metabolic syndrome”.
Lines 51-136: While the authors have justified the inclusion of factors in the development of metabolic syndrome and hypertension in metabolic syndrome, the authors failed to make relevant connection for each of the factors in relation to how they affect hypertension in metabolic syndrome. The authors provide very general statements throughout the manuscript e.g. adipocytokines may contribute to the development of hypertension (line 56), without giving details on the underlying mechanistic role of the factors (i.e. obesity/adipocytokines) in contributing to hypertension in metabolic syndrome. Another example would be “Enhanced fructose consumption elicits the development of chronic low-grade inflammation and expansion of white adipose tissue in various models of metabolic syndrome (lines 54-56)”, how is this related in the context of obesity in contributing to hypertension. This statement of introducing fructose seems out of place. In addition, the authors listed “salt overload” as one of the factors in the development of hypertension in metabolic syndrome but the review is on role of salt in pathogenesis of hypertension in metabolic syndrome so there might be overlap in this context.
Lines 137-167: This section should be under the obesity subsection? How is this section different from stating that obesity is a factor contributing to hypertension in metabolic syndrome (lines 51-57)? The authors mentioned that “Among these factors, visceral obesity has been proposed as a key player in the development of hypertension (lines 48-49)” but they did not provide further elaboration on how visceral adiposity is involved in the development of hypertension.
Lines 173-192: This section is on “The Role of Excessive Carbohydrate and Salt Consumption in the development of Hypertension and the onset of metabolic syndrome” but how did the authors define “onset of metabolic syndrome”-there is no indication of this onset in the write-up? How is this section different from the section “The role of enhanced salt and carbohydrate consumption in the pathogenesis of hypertension in metabolic syndrome (lines 238-239)”?
Lines 225-237: The section on “Stimulatory effect of excessive carbohydrate consumption on salt absorption in the small intestine” can be combined with lines 250-302 which also focus on salt absorption in the small intestine.
Lines 339-370: In the author’s response to the reviewer’s comments, the authors mentioned that “increased production of uric acid, as well as alteration of gut microbiome, is consequent to increased dietary fructose-containing food”. However, how increased intake of fructose-containing food affect gut microbiome was not described. The authors also provide very general statement “Fructose consumption raises serum levels of uric acid” without providing mechanistic explanation. Also, what is the relation of western diet and metabolic syndrome in the context of fructose and salt in the development of hypertension in metabolic syndrome?
Author Response
1/31/2023
Reviewer 2 (Round 2)
Thank you for the opportunity to review the revised manuscript. The authors have made some amendments. However, most of the responses did not directly address the reviewer’s main concerns. The structure of the manuscript remains disorganized and it lacks details and description on important points.
General comments: It remains unclear if the focus of the review is on role of fructose and salt on hypertension in metabolic syndrome or is it on “the role of fructose in stimulating salt absorption and its impact on the development of hypertension in metabolic syndrome” as given as response to the reviewer’s comment. If it is the latter, please kindly revise the title and structure of the manuscript accordingly.
Response: We had revised the title as requested by the referee during the first round. The referee had indicated 2 possible options for the title (options A and B). We had indicated in our response that we had selected option B, which is the current title.
Related to the concern of referee 2, we are stating that fructose-stimulated salt absorption in the small intestine and kidney tubules will lead to a state of salt overload and eventual hypertension. We believe that the above statement is encompassing of the role of fructose and salt on the genesis of hypertension in metabolic syndrome. Our detailed graphics in Figs. 1, 3, 4 and 5 are all supportive of either the direct effect of fructose or fructose-mediated insulin resistance on enhanced salt absorption in the small intestine and kidney.
- Line 46: Please change “this condition” to “metabolic syndrome”.
Response: We have changed “this condition” to “metabolic syndrome” as requested.
- Lines 51-136: While the authors have justified the inclusion of factors in the development of metabolic syndrome and hypertension in metabolic syndrome, the authors failed to make relevant connection for each of the factors in relation to how they affect hypertension in metabolic syndrome. The authors provide very general statements throughout the manuscript e.g. adipocytokines may contribute to the development of hypertension (line 56), without giving details on the underlying mechanistic role of the factors (i.e. obesity/adipocytokines) in contributing to hypertension in metabolic syndrome. Another example would be “Enhanced fructose consumption elicits the development of chronic low-grade inflammation and expansion of white adipose tissue in various models of metabolic syndrome (lines 54-56)”, how is this related in the context of obesity in contributing to hypertension. This statement of introducing fructose seems out of place. In addition, the authors listed “salt overload” as one of the factors in the development of hypertension in metabolic syndrome but the review is on role of salt in pathogenesis of hypertension in metabolic syndrome so there might be overlap in this context.
Response: The role of fructose and salt in obesity and insulin resistance has been established and the references are cited in this manuscript (20-22). Furthermore, the contribution of both fructose and salt to obesity and adipocytokine elaboration is documented and the references are cited in the manuscript (20-22). In addition, adipocytokines such as MCP-1, TNF-a, IL-6 and IL-17a enhance renal sodium retention and alter vascular tonicity leading to the development of hypertension. New references to this effect are now included (20-22). Taken as a whole, a picture emerges where increased salt and fructose intake contribute to the development of multiple determinants of metabolic syndrome, including insulin resistance, obesity, and hypertension. A statement regarding this issue as well as relevant references is now included in the manuscript (lines 60-65). We hope that this statement and the inclusion of new references address the concerns of referee 2.
- Lines 137-167: This section should be under the obesity subsection? How is this section different from stating that obesity is a factor contributing to hypertension in metabolic syndrome (lines 51-57)? The authors mentioned that “Among these factors, visceral obesity has been proposed as a key player in the development of hypertension (lines 48-49)” but they did not provide further elaboration on how visceral adiposity is involved in the development of hypertension.
Response: This review article is divided into 2 main sections. The first section encompasses lines 31 to 180 and discusses the increased prevalence of metabolic syndrome and the main contributing factors that are deemed critical to the generation of hypertension in this rapidly escalating condition. The last section covers lines 181 to 337 and focuses on the specific roles of fructose and salt and their mechanistic basis in the generation of hypertension in metabolic syndrome.
As stated in response to Concern #2, we have now provided more information on the role of visceral obesity, low grade inflammation, and salt sensitive hypertension in metabolic syndrome, and specifically in fructose and salt-mediated hypertension (20-22).
- Lines 173-192: This section is on “The Role of Excessive Carbohydrate and Salt Consumption in the development of Hypertension and the onset of metabolic syndrome” but how did the authors define “onset of metabolic syndrome”-there is no indication of this onset in the write-up? How is this section different from the section “The role of enhanced salt and carbohydrate consumption in the pathogenesis of hypertension in metabolic syndrome (lines 238-239)”?
Response: We have deleted the word “the onset” and the statement now reads as follows “The Role of Excessive Carbohydrate and Salt Consumption in the Development of Metabolic Syndrome and Hypertension”.
- Lines 225-237: The section on “Stimulatory effect of excessive carbohydrate consumption on salt absorption in the small intestine” can be combined with lines 250-302 which also focus on salt absorption in the small intestine.
Response: We agree. We have now moved the subsection titled “Stimulatory effect of excessive carbohydrate consumption on salt absorption in the small intestine” to the section titled, “The role of enhanced salt and carbohydrate consumption in the pathogenesis of hypertension in metabolic syndrome”. This is followed by a new title “Molecules mediating fructose-stimulated salt absorption in the small intestine” in the following subsection. We hope this new restructuring will address the concern of the referee.
- Lines 339-370: In the author’s response to the reviewer’s comments, the authors mentioned that “increased production of uric acid, as well as alteration of gut microbiome, is consequent to increased dietary fructose-containing food”. However, how increased intake of fructose-containing food affect gut microbiome was not described. The authors also provide very general statement “Fructose consumption raises serum levels of uric acid” without providing mechanistic explanation. Also, what is the relation of western diet and metabolic syndrome in the context of fructose and salt in the development of hypertension in metabolic syndrome?
Response: We have included appropriate references for sections discussing the role of enhanced fructose consumption in increasing uric acid production (Refs. 123-124) and alteration in gut microbiome (Refs. 4, 125 and 126), and their impact on blood pressure elevation. Due to space constraint (we have over 7700 words in the manuscript), we feel the inclusion of those references is adequate to highlight the role and basis of elevated uric acid and altered gut microbiome in the pathogenesis of hypertension in metabolic syndrome.
In regards to the Western diet, it is important to note that recent studies have indicated that diets mimicking the Western diet (high fructose, high fat, high salt) may have a more significant impact on the magnitude and/or the onset of the development of hypertension in animal models. Appropriate references are included (Refs. 127-130). This is a new area and for sure requires more investigation.

Reviewer 3 Report
Dear authors, I appreciate your response to my criticism and accept your answers, and the changes to the manuscript you made.
Author Response
We thank the referee for careful review and positive assessment of our manuscript.